# Automatic differentiation of nonsmooth iterative algorithms

**Jérôme Bolte**
Toulouse School of Economics,
University of Toulouse Capitole.
Toulouse, France.

**Edouard Pauwels**
IRIT, CNRS, Université de Toulouse.
Institut Universitaire de France (IUF).
Toulouse, France.

**Samuel Vaiter**
CNRS & Université Côte d'Azur,
Laboratoire J. A. Dieudonné.
Nice, France.

## Abstract

Differentiation along algorithms, i.e., piggyback propagation of derivatives, is now routinely used to differentiate iterative solvers in differentiable programming. Asymptotics is well understood for many smooth problems but the nondifferentiable case is hardly considered. Is there a limiting object for nonsmooth piggyback automatic differentiation (AD)? Does it have any variational meaning and can it be used effectively in machine learning? Is there a connection with classical derivative? All these questions are addressed under appropriate nonexpansivity conditions in the framework of conservative derivatives which has proved useful in understanding nonsmooth AD. We characterize the attractor set of nonsmooth piggyback iterations as a set-valued fixed point which remains in the conservative framework. Among various consequences we have almost everywhere convergence of classical derivatives. Our results are illustrated on parametric convex optimization with forward-backward, Douglas-Rachford and Alternating Direction of Multiplier algorithms as well as the Heavy-Ball method.

## 1 Introduction

**Differentiable programming.** We consider a Lipschitz function $F: \mathbb{R}^p \times \mathbb{R}^m \mapsto \mathbb{R}^p$, representing an iterative algorithm, parameterized by $\theta \in \mathbb{R}^m$, with Lipschitz initialization $x_0: \theta \mapsto x_0(\theta)$ and

$$x_{k+1}(\theta) = F(x_k(\theta), \theta) = F_\theta(x_k(\theta)), \tag{1}$$

where $F_\theta := F(\cdot, \theta)$, under the assumption that $x_k(\theta)$ converges to the unique fixed point of $F_\theta$: $\bar{x}(\theta) = \text{fix}(F_\theta)$. Such recursion represent for instance algorithms to solve an optimization problem $\min_x h(x)$ (e.g. empirical risk minimization), such as gradient descent: $F(x, \theta) = x - \theta \nabla h(x)$. But (1) could also be a fixed-point equation such as a deep equilibrium network [5].

Figure 1: We study existence and meaning of $J_{\bar{x}}^{\text{pb}}$ as a derivative of $\bar{x}$, compatible with automatic differentiation of the iterates $(x_k(\theta))_{k \in \mathbb{N}}$.

In the last years, a paradigm shift occurred: such algorithms are now implemented in algorithmic differentiation (AD)-friendly frameworks such as Tensorflow [1], PyTorch [42] or JAX [13]. For a differentiable $F$, it is possible to compute iteratively the derivatives of

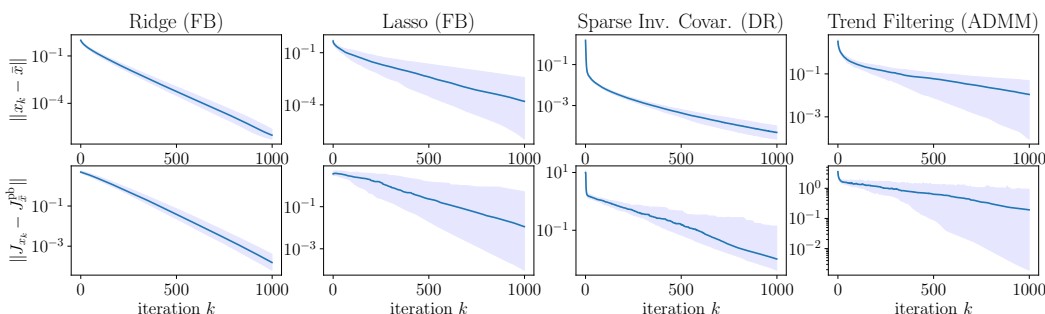

Figure 2: Illustration of the linear convergence of proximal splitting methods. *(First line)* Distance of the iterates to the fixed point. *(Second line)* Distance of the piggyback Jacobians to the Jacobian of the fixed point. The acronyms are FB for Forward-Backward, DR for Douglas-Rachford and ADMM for Alternating Direction Method of Multipliers. In all cases, despite nonsmoothness, piggyback Jacobians converge, illustrating Corollary 2. Blue lines represent the median of 100 repetitions with random data, and the blue shaded area represents the first and last deciles.

$x_k$ with respect to $\theta$ using the differential calculus rules resulting in so called "piggyback" recursion:

$$\frac{\partial}{\partial \theta} x_{k+1}(\theta) = \partial_1 F(x_k(\theta), \theta) \cdot \frac{\partial}{\partial \theta} x_k(\theta) + \partial_2 F(x_k(\theta), \theta), \tag{2}$$

where $\frac{\partial}{\partial \theta} x_k$ is the Jacobian of $x_k$ with respect to $\theta$. In practice, automatic differentiation frameworks do not compute the full Jacobian, but compute either vector-Jacobian products in reverse-mode (or backpropagation) [48] or Jacobian-vector products in forward mode [53]. We rather consider the full Jacobian, and therefore, our findings *apply to both* modes. We focus on two issues arising with nonsmooth recursions, illustrated in Figure 1. *(i)* what can be said about the chain rule (2) and its asymptotics when the function $F$ is not smooth (for example a projected gradient step)? *(ii)* how to interpret its asymptotics as a notion of derivative for $\bar{x}$, the fixed point of $F_\theta$? We propose a *joint* answer to both questions, providing a solid theoretical ground to the idea of algorithmic differentiation of numerical solvers involving nonsmooth components in a differentiable programming context.

**Related works.** Algorithmic use of the chain rule (2) to differentiate programs takes its root in [53], with forward differentiation, and later in reverse mode [35]. Along with the development of AD, convergence of the iterative sequence (2) was investigated, notably in the optimization community as reviewed in [28]. This important survey paper gathers results in differentiable programming rediscovered/reused later: implicit differentiation [43, 45] and its stability [8], adjoint fixed point iteration [5] (a key aspect of the deep equilibrium network) and linear convergence of (2). Notably, linear convergence of Jacobians was investigated in [25, 27] for the forward mode and in [15, Theorem 2.3] for the reverse mode. This was more recently investigated – for $C^2$ *functions* – in imaging for primal-dual algorithms [14, 9] and in machine learning for gradient descent [39, 36] and the Heavy-ball [39] method. In the specific context where $F$ solves a min-min problem, the authors in [2] proved the linear convergence of the Jacobians. The use of AD for nonsmooth functions was justified with the notion of *conservative Jacobians* [12, 11] with a nonsmooth version of the chain rule for compositional models. Correctness of AD was also investigated in [34] for a large class of piecewise analytic functions, and in [33] where a qualification condition is used to compute a Clarke Jacobian. Along with AD, a natural way to differentiate a model (1) is by implicit differentiation, recently applied in several works [5, 3, 21]. In a nonsmooth context, an implicit function theorem [10] was proved for path-differentiable functions. In terms of applications, nonsmooth piggyback derivatives are applied to hyperparameter tuning for inverse problems in [8] while the case of Lasso was investigated in [7]. Other relevant applications include plug-and-play denoising [32], parameter selection [19], bilevel programming [41]

**Contributions:** Under suitable nonexpansivity assumptions, our contributions are as follows.
• We address both questions illustrated in Figure 1 for nonsmooth recursions. Set-valued extensions of the piggyback recursion (2) have a well defined limit: the fixed point of subset map (Theorem 1), it is conservative for the fixed point map $\bar{x}$. This is a nonsmooth "infinite" chain rule for AD (Theorem 2).

- For almost all $\theta$, despite nonsmoothness, recursion (2) is well defined using the classical Jacobian and converges to the classical Jacobian of the fixed point $\bar{x}$ (Corollary 2). This has implications for both forward and reverse modes of AD.
- For a large class of functions (Lipschitz-gradient selection), it is possible to give a quantitative rate estimate (Corollary 3), namely to prove linear convergence of the derivatives.
- We show that these results can be applied to proximal splitting algorithms in nonsmooth convex optimization. We include forward–backward (Proposition 2), as well Douglas–Rachford (Proposition 3) and ADMM, a numerical illustration of the convergence of derivatives is given in Figure 2.
- We also illustrate that, contrarily to the smooth case, nonsmooth piggy back derivatives of momentum methods such as Heavy-ball, may diverge even if the iterates converge linearly (Proposition 4).

**Notations.** A function $f : \mathbb{R}^p \to \mathbb{R}^m$ is locally Lipschtiz if, for each $x \in \mathbb{R}^n$, there exists a neighborhood of $x$ on which $f$ is Lipschitz. Denoting by $R \subseteq \mathbb{R}^p$, the full measure set where $f$ is differentiable, the Clarke Jacobian [16] at $x \in \mathbb{R}^p$ is defined as

$$\mathrm{Jac}\,^c f(x) = \mathrm{conv} \left\{ M \in \mathbb{R}^{p \times m}, \exists (y_k)_{k \geqslant 0} \text{ s.t. } \lim_{k \to \infty} y_k = x, y_k \in R, \lim_{k \to \infty} \frac{\partial f}{\partial y}(y_k) = M \right\}. \quad (3)$$

The Clarke subdifferential, $\partial^c f$ is defined similarly. Given two matrices $A, B$ with compatible dimension, $[A, B]$ is their concatenation. For a set $\mathcal{X}$, $\mathrm{conv}\mathcal{X}$ is its convex hull. The symbol $\mathbb{B}$ denotes a unit ball, the corresponding norm should be clear from the context.

## 2    Nonsmooth piggyback differentiation

We first show how the use of the notion of *conservative Jacobians* allow us to justify rigorously the existence of a nonsmooth equivalent of piggyback iterations in (2) that is compatible with AD.

**Conservative Jacobians.** Conservative Jacobians were introduced in [12] as a generalization of derivatives to study automatic differentiation of nonsmooth functions. Given a locally Lipschitz continuous function $f : \mathbb{R}^p \to \mathbb{R}^m$, the set-valued $J : \mathbb{R}^p \rightrightarrows \mathbb{R}^{m \times p}$ is a *conservative Jacobian* for the *path differentiable* $f$ if $J$ has a closed graph, is locally bounded and nowhere empty with

$$\frac{d}{dt} f(\gamma(t)) = J(\gamma(t))\dot{\gamma}(t) \quad \text{a.e.} \quad (4)$$

for any $\gamma \colon [0,1] \to \mathbb{R}^p$ absolutely continuous with respect to the Lebesgue measure. Conservative gradients are defined similarly. We refer to [12] for extensive examples and properties of this class of function, key ideas are recalled in Appendix A for completeness. Let us mention that the classes of convex functions, definable functions, or semialgebraic functions are contained in the set of path differentiable functions. Given $D_f : \mathbb{R}^p \rightrightarrows \mathbb{R}^p$, a conservative gradient for $f : \mathbb{R}^p \to \mathbb{R}$, we have:

- (**Clarke subgradient**), for all $x \in \mathbb{R}^p$, $\partial^c f(x) \subset \mathrm{conv}(D_f(x))$.
- (**Gradient almost everywhere**) $D_f(x) = \{\nabla f(x)\}$ for almost all $x \in \mathbb{R}^p$.
- (**Calculus**) differential calculus rules preserve conservativity, *e.g.* sum and compositions of conservative Jacobians are conservative Jacobians.

Finally, $D_f$ can be used as a first order optimization oracle for methods of gradient type [11].

**Piggyback differentiation of recursive algorithms.** The following is standing throughout the text.

**Assumption 1 (The conservative Jacobian of the iteration mapping is a contraction)** $F$ is locally Lipschitz, path differentiable, jointly in $(x, \theta)$, and $J_F$ is a conservative Jacobian for $F$. There exists $0 \leqslant \rho < 1$, such that for any $(x, \theta) \in \mathbb{R}^p \times \mathbb{R}^m$ and any pair $[A, B] \in J_F(x, \theta)$, with $A \in \mathbb{R}^{p \times p}$ and $B \in \mathbb{R}^{p \times m}$, the operator norm of $A$ is at most $\rho$. $J_{x_0}$ is a conservative Jacobian for the initialization function $\theta \mapsto x_0(\theta)$.

Under Assumption 1, $F_\theta$ is a strict contraction so that $(x_k(\theta))_{k \in \mathbb{N}}$ converges linearly to $\bar{x}(\theta) = \mathrm{fix}(F_\theta)$ the unique fixed point of the iteration mapping $F_\theta$. More precisely, for all $k \in \mathbb{N}$, we have

$$\|x_k(\theta) - \bar{x}(\theta)\| \leqslant \rho^k \frac{\|x_0 - F_\theta(x_0)\|}{1 - \rho}.$$

Furthermore, for every $k \in \mathbb{N}$, let us define the following set-valued piggyback recursion:

$$J_{x_{k+1}}(\theta) = \{AJ + B, \ [A,B] \in J_F(x_k(\theta), \theta), \ J \in J_{x_k}(\theta)\}. \tag{PB}$$

We will show in Section 3 that (PB) plays the same role as (2) in the nonsmooth setting. Note that one can recursively evaluates a sequence $J_k \in J_{x_k}$, $k \in \mathbb{N}$, through operations actually implemented in nonsmooth AD frameworks, as follows

$$J_{k+1} = A_k J_k + B_k \quad \text{where} \quad [A_k, B_k] \in J_F(x_k(\theta), \theta), \tag{5}$$

**Remark 1 (Local contractions)** Assumption 1 may be relaxed as follows: for all $\theta$, the fixed point set $\mathrm{fix}(F_\theta)$ is a singleton $\bar{x}_\theta$ such that $x_k(\theta) \to \bar{x}(\theta)$ as $k \to \infty$, and the operator norm condition on $J_F$ in Assumption 1 holds at the point $(\bar{x}(\theta), \theta)$. By graph closedness of $J_F$, in a neighborhood of $(\bar{x}(\theta), \theta)$, $F_\theta$ is a strict contraction and the operator norm condition on $J_F$ holds, possibly with a larger contraction factor $\rho$. After finitely many steps, the iterates $(x_k)_{k \in \mathbb{N}}$ remain on some neighborhood and all our convergence results hold, due to their asymptotic nature.

**Remark 2 (Relation to existing work)** For a smooth $F$ a natural conservative Jacobian is the classical one. The, hypotheses in [39, 36] for gradient descent ($F$ is $C^1$), are exactly the classical counterpart of Assumption 1. On the other hand [25, 27, 15] use a more general assumption on spectral radius, which allow to treat the Heavy-Ball method, *e.g.* in [39]. However this does not generalize to sets of matrices, as shown in Section 5. Hence Assumption 1 is on operator norm and not on spectral radius, which is sharp, contrary to the smooth case.

## 3 Asymptotics of nonsmooth piggyback differentiation

### 3.1 Fixed point of affine iterations

**Gap and Haussdorf distance.** Being given two nonempty compact subsets $\mathcal{X}, \mathcal{Y}$ of $\mathbb{R}^p$, set

$$\mathrm{gap}(\mathcal{X}, \mathcal{Y}) = \max_{x \in \mathcal{X}} d(x, \mathcal{Y}) \quad \text{where} \quad d(x, \mathcal{Y}) = \min_{y \in \mathcal{Y}} \|x - y\|,$$

and define the Hausdorff distance between $\mathcal{X}$ and $\mathcal{Y}$ by $\mathrm{dist}(\mathcal{X}, \mathcal{Y}) = \max(\mathrm{gap}(\mathcal{X}, \mathcal{Y}), \mathrm{gap}(\mathcal{Y}, \mathcal{X}))$. Note that $\mathrm{gap}(\mathcal{X}, \mathcal{Y}) = 0$ if, and only if, $\mathcal{X} \subseteq \mathcal{Y}$, whereas $\mathrm{dist}(\mathcal{X}, \mathcal{Y}) = 0$ if, and only if, $\mathcal{X} = \mathcal{Y}$. Moreover, $\mathcal{X} \subseteq \mathcal{Y} + \mathrm{gap}(\mathcal{X}, \mathcal{Y})\mathbb{B}$ where $\mathbb{B}$ is the unit ball. It means that $\mathrm{gap}(\mathcal{X}, \mathcal{Y})$ "measures" the default of inclusion of $\mathcal{X}$ in $\mathcal{Y}$, see [46, Chapter 4] for more details.

**Affine iterations by packets of matrices.** Let $\mathcal{J} \subset \mathbb{R}^{p \times (p+m)}$ be a compact subset of matrices such that any matrix of the form $[A, B] \in \mathcal{J}$ with $A \in \mathbb{R}^{p \times p}$ is such that $A$ has operator norm at most $\rho < 1$. We let $\mathcal{J}$ act naturally on the matrices of size $p \times m$ as follows $\mathcal{J} \colon \mathbb{R}^{p \times m} \rightrightarrows \mathbb{R}^{p \times m}$ the function from $\mathbb{R}^{p \times m}$ to subsets of $\mathbb{R}^{p \times m}$ which is defined for each $X \in \mathbb{R}^{p \times m}$ as follows: $\mathcal{J}(X) = \{AX + B, \ [A, B] \in \mathcal{J}\}$. This defines a set-valued map through, for any $\mathcal{X} \subset \mathbb{R}^{p \times m}$,

$$\mathcal{J}(\mathcal{X}) = \{AX + B, \ [A, B] \in \mathcal{J}, \ X \in \mathcal{X}\}. \tag{6}$$

Recursions of the form (PB) generate sequences $(\mathcal{X}_k)_{k \in \mathbb{N}}$ of subsets of $\mathbb{R}^{p \times m}$ satisfying

$$\mathcal{X}_{k+1} = \mathcal{J}(\mathcal{X}_k) \quad \forall k \in \mathbb{N}. \tag{7}$$

The following is an instance of the Banach–Picard theorem (whose proof is recalled in Appendix B).

**Theorem 1 (Set-valued affine contractions)** *Let $\mathcal{J} \subset \mathbb{R}^{p \times (p+m)}$ be a compact subset of matrices as above with $\rho < 1$. Then there is a unique nonempty compact set $\mathrm{fix}(\mathcal{J}) \subset \mathbb{R}^{p \times m}$ satisfying $\mathrm{fix}(\mathcal{J}) = \mathcal{J}(\mathrm{fix}(\mathcal{J}))$, where the action of $\mathcal{J}$ is given in (6).*

*Let $(\mathcal{X}_k)_{k \in \mathbb{N}}$ be a sequence of compact subsets of $\mathbb{R}^{p \times m}$, such that $\mathcal{X}_0 \neq \varnothing$, and satisfying the recursion (7). We have for all $k \in \mathbb{N}$*

$$\mathrm{dist}(\mathcal{X}_k, \mathrm{fix}(\mathcal{J})) \leqslant \rho^k \frac{\mathrm{dist}(\mathcal{X}_0, \mathcal{J}(\mathcal{X}_0))}{1 - \rho},$$

*where* $\mathrm{dist}$ *is the Hausdorff distance related to the Euclidean norm on $p \times m$ matrices.*

## 3.2 An infinite chain rule and its consequences

Define the following set-valued map based on the fix operator from Theorem 1,

$$J_{\bar{x}}^{\text{pb}} \colon \theta \rightrightarrows \text{fix}\left[J_F(\bar{x}(\theta), \theta)\right].$$

where $\bar{x}(\theta)$ is the unique fixed point of the algorithmic recursion. Since $\bar{x}(\theta) = \text{fix}(F_\theta)$, we have equivalently that $J_{\bar{x}}^{\text{pb}}$ is the fixed-point of the Jacobian at the fixed-point: $J_{\bar{x}}^{\text{pb}} \colon \theta \rightrightarrows \text{fix}\left[J_F(\text{fix}(F_\theta), \theta)\right]$. We have the following (proved in Appendix C) and a consequence from Theorem 1.

**Theorem 2 (A conservative mapping for the fixed point map)** *Under Assumption 1, $J_{\bar{x}}^{\text{pb}}$ is well-defined, and is a conservative Jacobian for the fixed point map $\bar{x}$.*

**Corollary 1 (Convergence of the piggyback derivatives)** *Under Assumption 1, for all $\theta$, the recursion (PB) satisfies*

$$\lim_{k \to \infty} \text{gap}(J_{x_k}(\theta), J_{\bar{x}}^{\text{pb}}(\theta)) = 0. \tag{8}$$

Unrolling the expression of $J_{x_k}$, using (6) and (7), we can rewrite (8) with a compositional product:

$$\lim_{K \to +\infty} \text{gap}\left(\left(\bigcirc_{k=0}^{K} J_F(x_k(\theta), \theta)\right)(J_{x_0}(\theta)), J_{\bar{x}}^{\text{pb}}(\theta)\right) = 0.$$

In plain words, this is a limit-derivative exchange result: *Asymptotically, the gap between the automatic differentiation of $x_k$ and the derivative of the limit is zero.* In particular the recursion (5) produces bounded sequences whose accumulation points are in $J_{\bar{x}}^{\text{pb}}$. Since conservative Jacobians equal classical Jacobians almost everywhere [12], we have convergence of classical derivatives.

**Corollary 2 (Convergence a.e. of the classical piggyback derivatives)** *Under Assumption 1, for almost all $\theta$, the classical Jacobian $\frac{\partial}{\partial \theta} x_k(\theta)$, is well defined for all $k$ and converges towards the classical Jacobian of $\bar{x}$. That is*

$$\lim_{k \to \infty} \frac{\partial}{\partial \theta} x_k(\theta) = \frac{\partial}{\partial \theta} \bar{x}(\theta), \quad \text{for almost all } \theta.$$

**Remark 3 (Connection to implicit differentiation)** The authors in [10] proved a qualification-free version of the implicit function theorem. Assuming that for every $[A, B] \in J(\bar{x}(\theta), \theta)$, the matrix $I - A$ is invertible, we have that

$$J_{\bar{x}}^{\text{imp}} \colon \theta \rightrightarrows \left\{(I - A)^{-1} B, [A, B] \in J_F(\bar{x}(\theta), \theta)\right\} \tag{9}$$

is a conservative Jacobian for $\bar{x}$. Under Assumption 1, one has $J_{\bar{x}}^{\text{imp}}(\theta) \subset J_{\bar{x}}^{\text{pb}}(\theta)$ for all $\theta$. Unfortunately, as soon as $F$ is not differentiable, the inclusion may be strict, see details in Appendix D.

## 3.3 Consequence for algorithmic differentiation

Given $k \in \mathbb{N}$, $\dot{\theta} \in \mathbb{R}^m$, $\bar{w}_k \in \mathbb{R}^p$, the following algorithms allow us to compute $\dot{x}_k = J_k \dot{\theta}$ using the forward mode of automatic differentiation (Jacobian Vector Products, JVP), and $\bar{\theta}_k^T = \bar{w}_k^T J_k$ using the backward mode of automatic differentiation (Vector Jacobian Products, VJP). In a compositional model $\dot{\theta}$ is the derivative of an inner functions controlling algorithm parameters $\theta$, with another variable real variable $\lambda \in \mathbb{R}$, for example an hyper parameter. The goal is to combine $\frac{\partial \theta(\lambda)}{\partial \lambda}$ and $\frac{\partial x_k(\theta)}{\partial \theta}$ with the chain rule in a forward pass to obtain the total derivative $\frac{\partial x_k(\theta(\lambda))}{\partial \lambda}$. On the other hand, in a compositional model, $\bar{w}_k$ is typically the gradient of an outer loss functions $\ell$ evaluated at $x_k(\theta)$. In this case the goal is to combine derivatives of iterates $\frac{\partial x_k(\theta)}{\partial \theta}$ with $\bar{w}_k = \frac{\partial \ell(x_k)}{\partial x_k}$ in a backward pass to obtain $\frac{\partial \ell(x_k(\theta))}{\partial \theta}$.

**Algorithm 1:** Algorithmic differentiation of recursion (1), forward and reverse modes

---

**Input:** $k \in \mathbb{N}$, $\theta \in \mathbb{R}^m$, $\dot{\theta} \in \mathbb{R}^m$, $\bar{w}_k \in \mathbb{R}^p$, initialization function $x_0(\theta)$, recursion function $F(x,\theta)$, conservative Jacobians $J_F(x,\theta)$ and $J_{x_0}(\theta)$. Initialize: $x_0 = x_0(\theta) \in \mathbb{R}^p$.

**Forward mode (JVP):**
$\dot{x}_0 = J\dot{\theta}$, $J \in J_{x_0}(\theta)$.
**for** $i = 1, \ldots, k$ **do**
   $x_i = F(x_{i-1}, \theta)$
   $\dot{x}_i = A_{i-1}\dot{x}_{i-1} + B_{i-1}\dot{\theta}$
   $[A_{i-1}, B_{i-1}] \in J_F(x_{i-1}, \theta)$
**Return:** $\dot{x}_k$

**Reverse mode (VJP):** $\bar{\theta}_k = 0$.
**for** $i = 1, \ldots, k$ **do**
   $x_i = F(x_{i-1}, \theta)$
**for** $i = k, \ldots, 1$ **do**
   $\bar{\theta}_k = \bar{\theta}_k + B_{i-1}^T \bar{w}_i$   $\bar{w}_{i-1} = A_{i-1}^T \bar{w}_i$
   $[A_{i-1}, B_{i-1}] \in J_F(x_{i-1}, \theta)$
$\bar{\theta}_k = \bar{\theta}_k + J^T \bar{w}_0$, $J \in J_{x_0}(\theta)$
**Return:** $\bar{\theta}_k$

---

**Proposition 1 (Convergence of VJP and JVP)** *Let $k \in \mathbb{N}$, $\dot{\theta} \in \mathbb{R}^m$, $\bar{w}_k \in \mathbb{R}^p$, $x_k \in \mathbb{R}^p$, $\dot{x}_k \in \mathbb{R}^p$, $\bar{\theta}_k^T \in \mathbb{R}^m$ be as in Agorithm 1 under Assumption 1. Then for almost all $\theta \in \mathbb{R}^m$, $\dot{x}_k \to \frac{\partial \bar{x}}{\partial \theta}\dot{\theta}$.*

*Assume furthermore that, as $k \to \infty$, $\bar{w}_k \to \bar{w}$ (for example, $\bar{w}_k = \nabla\ell(x_k)$ for a $C^1$ loss $\ell$), then for almost all $\theta \in \mathbb{R}^m$, $\bar{\theta}_k^T \to \bar{w}^T \frac{\partial \bar{x}}{\partial \theta}$.*

**Remark 4** In addition to Proposition 1, in both cases, for all $\theta$, all accumulation points of both $\dot{x}_k$ and $\bar{\theta}_k^T$ are elements of $J_{\bar{x}}^{\text{pb}}\dot{\theta}$ and $\bar{w}^T J_{\bar{x}}^{\text{pb}}$ respectively. This is a consequence of Corollary 2 combined with algorithmic differentiation arguments which proof is given in Appendix D.

### 3.4 Linear convergence rate for semialgebraic piecewise smooth selection function

Semialgebraic functions are ubiquitous in machine learning (piecewise polynomials, $\ell_1$, $\ell_2$ norms, determinant matrix rank . . . ). We refer the reader to [11] for a thorough discussion of their extensions, and use in machine learning. For more technical details, see [17, 18] for introductory material on semialgebraic and o-minimal geometry.

**Lipschitz gradient selection functions.** Let $F \colon \mathbb{R}^p \mapsto \mathbb{R}^q$ be semialgebraic and continuous. We say that $F$ has a *Lipschitz gradient selection* $(s, F_1, \ldots, F_m)$ if $s \colon \mathbb{R}^p \mapsto (1, \ldots, m)$ is semialgebraic and there exists $L \geqslant 0$ such that for $i = 1 \ldots, m$, $F_i \colon \mathbb{R}^p \mapsto \mathbb{R}^p$ is semial-gebraic with $L$-Lipschitz Jacobian, and for all $x \in \mathbb{R}^p$, $F(x) = F_{s(x)}(x)$. For any $x \in \mathbb{R}^p$, set $I(x) = \{i \in \{1, \ldots, m\}, F(x) = F_i(x)\}$. The set-valued map $J_F^s \colon \mathbb{R}^p \rightrightarrows \mathbb{R}^{p \times q}$ given by $J_F^s \colon x \rightrightarrows \text{conv}\left(\left\{\frac{\partial F_i}{\partial x}(x), i \in I(x)\right\}\right)$, is a conservative Jacobian for $F$ as shown in [11]. Here $\frac{\partial F_i}{\partial x}$ denotes the classical Jacobian of $F_i$. Let us stress that such a structure is ubiquitous in applications [11, 34].

**Rate of convergence.** We may now strengthen Corollary 1 by proving the linear convergence of piggyback derivatives towards the fixed point. The following is a consequence of the fact that the proposed selection conservative Jacobians of Lipschitz gradient selection functions are Lipschitz-like (Lemma 4 in Appendix E.1). Note that semialgebraicity is only used as a *sufficient* condition to ensure conservativity of the selection Jacobian together with this Lipschitz like property. It could be relaxed if it can be guaranteed by other means, in particular one could consider the broader class of definable functions in order to handle log-likelihood data fitting terms.

**Corollary 3 (Linear convergence of piggyback derivatives)** *In addition to Assumption 1, assume that $F$ has a Lipschitz gradient selection structure as above. Then, for any $\theta$ and $\epsilon > 0$, there exists $C > 0$ such that the recursion (PB) with $J_F = J_F^s$ satisfies for all $k \in \mathbb{N}$, $\text{gap}(J_{x_k}(\theta), J_{\bar{x}}^{\text{pb}}(\theta)) \leqslant C(\sqrt{\rho} + \epsilon)^k$. Moreover, classical Jacobians in Corollary 2 converge at a linear rate for almost all $\theta$.*

## 4 Application to proximal splitting methods in convex optimization

Consider the composite parametric convex optimization problem, where $\theta \in \mathbb{R}^m$ represents parameters and $x \in \mathbb{R}^p$ is the decision variable

$$\bar{x}(\theta) = \arg\min_x f(x, \theta) + g(x, \theta).$$

The purpose of this section is to construct examples of functions $F$ used in recursion (1) based on known algorithms. The following assumption will be standing throughout the section.

**Assumption 2** $f$ is semialgebraic, convex, its gradient with respect to $x$ for fixed $\theta$, $\nabla_x f$, is locally Lipschitz jointly in $(x, \theta)$ and $L$-Lipschitz in $x$ for fixed $\theta$. Semialgebraicity implies that $\nabla_x f$ is path-differentiable jointly in $(x, \theta)$, we denote by $J_f^2$ its Clarke Jacobian. The function $g$ is semialgebraic, convex in $x$ for fixed $\theta$, and lower semicontinuous. For all $\alpha > 0$, we assume that $G_\alpha(x, \theta) \mapsto \text{prox}_{\alpha g(\cdot, \theta)}(x)$ is locally Lipschitz jointly in $(x, \theta)$. Semialgebraicity implies that it is also path differentiable jointly in $(x, \theta)$, we denote by $J_{G_\alpha}$ its Clarke Jacobian.

This assumption covers a very large diversity of problems in convex optimization as most gradient and prox operations used in practice are semialgebraic (or definable). Under Assumption 2, we will provide sufficient conditions on $f$ and $g$ for Assumption 1, for different algorithmic recursions. These will therefore imply convergence as stated in Corollary 1 and 2, Proposition 1, as well Corollary 3 in the piecewise selection case. The proofs are postponed to Appendix F.

## 4.1 Splitting algorithms

In this section we provide sufficient condition for Assumption 1 to hold. The underlying conservative Jacobian is obtained by combining Clarke Jacobians of elementary algorithmic operations (gradient, proximal operator in Assumption 2), using the compositional rules of differential calculus [11] and implicit differentiation [10]. Using [12], such Jacobians are conservative by semialgebraicity and their combination provide conservative Jacobians for the corresponding algorithmic recursion $F$. These objects are explicitly constructed in Appendix F.

**Forward–backward algorithm.** The forward–backward iterations are given for $\alpha > 0$ by

$$x_{k+1} = \text{prox}_{\alpha g(\cdot, \theta)}\left(x_k - \alpha \nabla_x f(x_k, \theta)\right). \tag{10}$$

**Proposition 2** *Under Assumption 2 with $0 < \alpha < \frac{2}{L}$, denote by $F_\alpha \colon \mathbb{R}^{p \times m} \to \mathbb{R}^p$ the forward-backward recursion in (10). For $\mu > 0$, if either $f$ or $g$ is $\mu$-strongly convex in $x$ for all $\theta$, then $F_\alpha$ is a strict contraction and Assumption 1 holds.*

It is well known that if $f$ is $\mu$-strongly convex, choosing $\alpha = 2/(L + \mu)$ provides a contraction factor $\rho_{FB} = (\tau - 1)/(1 + \tau)$, where $\tau = L/\mu \geqslant 1$ is the condition number of the problem.

**Douglas–Rachford algorithm.** Given $\alpha > 0$, the algorithm goes as follows

$$y_{k+1} = \frac{1}{2}(I + R_{\alpha f(\cdot, \theta)} R_{\alpha g(\cdot, \theta)})y_k, \tag{11}$$

where $R_{\alpha f(\cdot, \theta)} = 2\text{prox}_{\alpha f(\cdot, \theta)} - I$ is the reflected proximal operator, which is 1-Lipschitz (and similarly for $g$). Following [6, Theorem 26.11], if the problem has a minimizer, then $(y_k)_{k \in \mathbb{N}}$ converges to a fixed point of (11), $\bar{y}$ such that $\bar{x} = \text{prox}_{\alpha g}(\bar{y})$ is a solution to the optimization problem. Following [26, Theorem 1], if $f$ is strongly convex, then $R_{\alpha f(\cdot, \theta)}$ is $\rho$-Lipschitz for some $\rho < 1$ and our differentiation result applies to Douglas-Rachford splitting in this setting.

**Proposition 3** *Under Assumption 2 with $\alpha > 0$, denote by $F_\alpha \colon \mathbb{R}^{p \times m} \to \mathbb{R}^p$ the Douglas-Rachford recursion in (11). If $f$ is $\mu$-strongly convex in $x$ for all $\theta$, then $F_\alpha$ is a strict contraction and Assumption 1 holds.*

Following [26, Proposition 3], choosing $\alpha = 1/\sqrt{L\mu}$ provides a contraction factor of order $\rho_{DR}(\sqrt{\tau} - 1)/(\sqrt{\tau} + 1) < \rho_{FB}$, where again $\tau = L/\mu$ is the condition number of the problem. In this respect Douglas-Rachford's iterations provide a faster asymptotic rate than those of Forward-Backward, which may also impact the convergence of derivatives in the context of Corollary 3.

**Alternating Direction Method of Multipliers.** Consider the separable convex problem

$$\min_{u,v} \phi_\theta(u) + \psi_\theta(v) \quad \text{subject to} \quad A_\theta u + B_\theta v = c_\theta. \tag{12}$$

The alternating direction method of multipliers (ADMM) algorithm combines two partial minimization of an augmented Lagrangian, and a dual update:

$$u_{k+1} = \arg\min_u \left\{ \phi_\theta(u) + x_k^\top A_\theta u + \frac{\alpha}{2} \|A_\theta u + B_\theta v_k - c_\theta\|_2^2 \right\}$$

$$v_{k+1} = \arg\min_v \left\{ \psi_\theta(v) + x_k^\top B_\theta v + \frac{\alpha}{2} \|A_\theta u_{k+1} + B_\theta v_k - c_\theta\|_2^2 \right\} \tag{13}$$

$$x_{k+1} = x_k + \alpha(A_\theta u_{k+1} + B_\theta v_{k+1} - c_\theta).$$

As observed in [23], the ADMM algorithm can be seen as the Douglas-Rachford splitting method applied to the Fenchel dual of problem (12) (see Appendix F.3 for more details). More precisely, ADMM updates are equivalent to Douglas-Rachford iterations applied to the following problem

$$\min_x \underbrace{c_\theta^\top x + \phi_\theta^*(-A_\theta^\top x)}_{f(x,\theta)} + \underbrace{\psi_\theta^*(-B_\theta^\top x)}_{g(x,\theta)} . \tag{14}$$

Therefore, if $\phi_\theta$ is strongly convex with Lipschitz gradient and $A_\theta$ is injective, then ADMM converges linearly and one is able to combine derivatives of proximal operators to differentiate ADMM.

## 4.2 Numerical illustrations

We now detail how Figure 2 discussed in the introduction is obtained, and how it illustrates our theoretical results. We consider four scenarios (Ridge, Lasso, Sparse inverse covariance selection and Trend filtering) corresponding to the four columns. For each of them, the first line shows the empirical linear rate of the iterates $x_k$ and the second line shows the empirical linear rate of the derivative $\frac{\partial}{\partial \theta} x_k$. All experiments are repeated 100 times and we report the median along with the first and last deciles.

**Forward–Backward for the Ridge.** The Ridge estimator is defined for $\theta > 0$ as $\bar{x}(\theta) = \arg\min_{x \in \mathbb{R}^p} \frac{1}{2}\|Ax - b\|_2^2 + \theta\|x\|_2^2$ Among several possibilities to solve it, one can use the Forward–Backward algorithm applied to $f \colon (x, \theta) \mapsto \frac{1}{2}\|Ax - b\|_2^2$ and $g \colon \theta\|x\|_2^2$. Since $g$ is strongly convex, the operator $F_\alpha$ is strongly convex, and thus Proposition 2 may be applied.

**Forward–Backward algorithm for the Lasso.** Consider the Forward–Backward algorithm applied to the Lasso problem [49], with parameter $\theta > 0$, $\bar{x}(\theta) \in \arg\min_{x \in \mathbb{R}^p} \frac{1}{2}\|Ax - b\|_2^2 + \theta\|x\|_1 = \arg\min_x \frac{1}{2L}\|Ax - b\|_2^2 + \frac{\theta}{L}\|x\|_1$, where $L$ is any upper bound on the operator norm of $A^T A$. The gradient of the quadratic part is 1-Lipschitz, so we may consider the forward backward algorithm (10), with unit step size and $f \colon (x, \theta) \mapsto \frac{1}{2L}\|Ax - b\|_2^2$ and $g \colon (x, \theta) \mapsto \frac{\theta}{L}\|x\|_1$.

A well known qualification condition involving a generalized support at optimality ensures uniqueness of the Lasso solution [20, 37]. It holds for generic problem data [50]. Following [10, Proposition 5], under this qualification condition, the implicit conservative Jacobian $J_F$ is such that, at the solution $x^*$, the matrix set $I - J_F$ only contains invertible matrices. This means that there exists $\rho < 1$, such that any $M \in J_F(x^*)$ has operator norm at most $\rho$. Following Remark 1, all our convergence results apply qualitatively. Note that we recover the results of [7, Proposition 2] for the Lasso.

**Douglas–Rachford for the Sparse Inverse Covariance Selection.** The Sparse Inverse Covariance Selection [52, 22] reads $\bar{x}(\theta) \in \arg\min_{x \in \mathbb{R}^{n \times n}} \text{tr}(Cx) - \log \det x + \theta \sum_{i,j} |x_{i,j}|$, where $C$ is a symmetric positive matrix and $\theta > 0$. It is possible to apply Douglas–Rachford method to $f \colon (x, \theta) \mapsto \text{tr}(Cx) - \log \det x$ and $g \colon (x, \theta) \mapsto \theta\|x\|_{1,1}$. It is known that $f$ is locally strongly convex, indeed $x \mapsto -\log \det x$ is the standard self-concordant barrier in semidefinite programming [40]. Following Remark 1, all our convergence results apply qualitatively.

**ADMM for Trend Filtering.** Introduced in [51] in statistics as a generalization of the Total Variation, the trend filtering estimator with observation $\theta \in \mathbb{R}^p$ reads $\bar{x}(\theta) = \arg\min_{x \in \mathbb{R}^p} \frac{1}{2}\|x - \theta\|_2^2 + \lambda\|D^{(k)}x\|_1$, where $D^{(k)}$ is a forward finite–difference approximation of a differential operator of order $k$ (here $k = 2$). Using $\psi_\theta \colon u \mapsto \lambda\|u\|_1$, $\phi_\theta \colon v \mapsto \|v - \theta\|_2^2$ (strongly convex), $A_\theta = -I$ (injective), $B_\theta = D^{(k)}$, and $c_\theta = 0$, we can apply the ADMM to solve trend filtering.

# 5 Failure of automatic differentiation for inertial methods

This section focuses on the Heavy-Ball method for strongly convex objectives, in its global linear convergence regime. For $C^2$ objectives, piggyback derivatives converge to the derivative of the solution map [28, 39, 36]. However, we provide a $C^{1,1}$ strongly convex parametric objective with path differentiable derivative, such that piggyback derivatives of the Heavy Ball algorithm contain diverging vectors for a given parameter value. In this example, other conservative differentiation means (implicit differentiation, piggyback on gradient descent), avoid this divergent behaviors.

## 5.1 Heavy-ball algorithm and global convergence

Consider a function $f\colon \mathbb{R}^p \times \mathbb{R}^m \to \mathbb{R}$, and $\beta > 0$, for simplicity, when the second argument is fixed, we write $f_\theta\colon x \mapsto f(x,\theta)$. Set for all $x,y,\theta$, $F(x,y,\theta) = (x - \nabla f_\theta(x) + \beta(x-y), x)$, consider the Heavy-Ball algorithm $(x_{k+1}, y_{k+1}) = F(x_k, y_k, \theta)$ for $k \in \mathbb{N}$.

If $f_\theta$ is $\mu$-strongly convex with $L$-Lipschitz gradient, then, choosing $\alpha = 1/L$ and $\beta < \frac{1}{2}\left(\frac{\mu}{2L} + \sqrt{\frac{\mu^2}{4L^2} + 2}\right)$, the algorithm will converge globally at a linear rate to the unique solution, $\bar{x}(\theta)$ [24, Theorem 4], local convergence is due to Polyak [44]. Furthermore, if in addition $f$ is $C^2$ forward propagation of derivatives converge to the derivative of the solution [28, 29, 39].

## 5.2 A diverging Jacobian accumulation

Details and proof of the following result are given in Section G.

**Proposition 4 (Piggyback differentiation fails for the Heavy Ball method)** *Consider $f\colon \mathbb{R}^2 \to \mathbb{R}$, such that for all $\theta \in \mathbb{R}$, $f(x,\theta) = x^2/2$ if $x \geqslant 0$ and $f(x,\theta) = x^2/8$ if $x < 0$. Assume that $\alpha = 1$ and $\beta = 3/4$. Then the heavy ball algorithm converges globally to $0$ and $\nabla f$ is path differentiable. The Clarke Jacobian of $F$ with respect to $(x,y)$ at $(0,0,0)$ is $J_F(0,0,0) = \mathrm{conv}\{M_1, M_2\}$, where the product $M_1 M_1 M_2 M_2$ has eigenvalue $-9/8$.*

The presence of an eigenvalue with modulus greater than 1 may produce divergence in (PB). Set

$$f_1\colon (x,\theta) \mapsto \begin{cases} x^2/2 & \text{if } x \geqslant 0 \\ x^2/8 & \text{if } x < 0. \end{cases} \qquad f_2\colon (x,\theta) \mapsto \begin{cases} x^2/2 & \text{if } x > 0 \\ x^2/8 & \text{if } x \leqslant 0. \end{cases}$$

Note that $f_1$ and $f_2$ are both equivalent to $f$ as they implement the same function. With initializations $x(\theta) = y(\theta) = \theta$, we run a few iterations of the Heavy Ball algorithm for $\theta = 0$, and implement (PB) alternating between two steps on $f_1$ and two steps on $f_2$ and differentiate the resulting sequence $(x_k)_{k \in \mathbb{N}}$ with respect to $\theta$ using algorithmic differentiation. The divergence phenomenon predicted by Proposition 4 is illustrated in Figure 3, while the true derivative is 0 (the sequence is constant).

# 6 Conclusion

We have developed a flexible theoretical framework to describe convergence of piggyback differentiation applied to nonsmooth recursions – providing, in particular, a rigorous meaning to differentiation of nonsmooth solvers. The relevance of our approach is illustrated on composite convex optimization

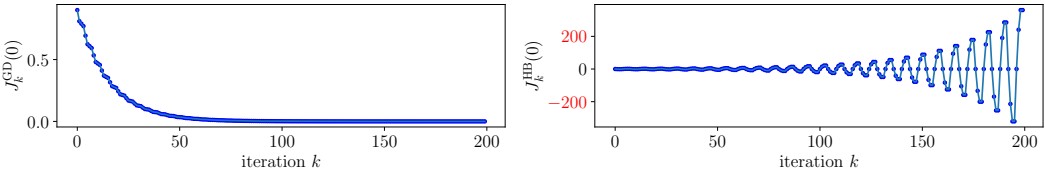

Figure 3: Behaviour of automatic differentiation for first-order methods on a quadratic function. (Left) Stability of the propagation of derivatives for the fixed step-size gradient descent. (Right) Instability of the propagation of Heavy-Ball initialized. Both methods are initialized at optimum.

through widely used methods as forward-backward, Douglas-Rachford or ADMM algorithms. Our framework allows however to consider many other abstract algorithmic recursions and provides thus theoretical ground for more general problems such as variational inequalities or saddle point problems as in [14, 9]. As a matter for future work, we shall consider relaxing Assumption 1 to study a wider class of methods, e.g., when $F$ is not a strict contraction.

## Acknowledgments

J. B. and E. P. acknowledge the financial support of the AI Interdisciplinary Institute ANITI funding under the grant agreement ANR-19-PI3A-0004, Air Force Office of Scientific Research, Air Force Material Command, USAF, under grant numbers FA9550-19-1-7026, FA8655-22-1-7012 and ANR MaSDOL 19-CE23-0017-01. J. B. also acknowledges the support of ANR Chess, grant ANR-17-EURE-0010, TSE-P and the Centre Lagrange. S. V. acknowledges the support ANR GraVa, grant ANR-18-CE40-0005.

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
