# OpenReview forum: "Automatic differentiation of nonsmooth iterative algorithms"
_NeurIPS.cc/2022/Conference — NeurIPS 2022 Accept_

### Official Review · Reviewer_Sht3 · 2022-07-02

**Rating:** 7
**Confidence:** 2
**Soundness:** 4 excellent
**Presentation:** 3 good
**Contribution:** 3 good

**Summary:**

In this paper, the authors present a theoretical framework based on nonsmooth analysis concepts to analyze the asymptotic behavior of « piggyback » derivatives for nonsmooth iterations, i.e. how we can define the automatic differentiation of the iterates of unrolled iterations of an algorithm with nondifferential updates. Cases of practical interest include most popular proximal algorithms : proximal gradient descent, Douglas Rachford splitting and the ADMM. The authors show that a notion of set-valued Jacobians can be defined for these algorithms, and that they coincide with classical Jacobians almost everywhere. What is more, they show that, under suitable assumptions, the fixed point admits a derivative in this sence, and the derivatives of the iterates converge towards the derivatives of the fixed point. They prove, again under suitable assumptions, that there is a linear convergence of the derivatives towards those of the fixed point. They also show that this does not apply to other optimization methods with momentum, the given example being the heavy ball methods for which Jacobian iterates can become unstable for an objective that is not regular enough.

**Questions:**

- Is it possible to find a case of practical interest where the heavy ball algorithm (or other momentum algorithms) will fail ? I feel like, from the example given, than in most cases the cost functions are regular enough for this not to happen, but I would appreciate to be proven wrong.

- How hard would the analysis get in the case where there are multiple fixed points of F ? E.g. when minimizing a nonconvex function ? I understand the analysis would get much more complex, and is probably out of the scope of the paper, but this could have been discussed a bit.

**Limitations:**

The authors state clearly the assumptions they need for their framework and discuss their relevance and connections to practical cases. Some important cases are not covered by the analysis, such as multiple fixed points (useful e.g. for nonconvex bilevel optimization), but this remark is quite minor since results in this case are scarce (to my knowledge) in the literature, and the analysis probably gets much harder. The connection to implicit differentiation is drawn and is interesting but unfortunately not equivalent to the piggyback differentation presented here.


**Strengths And Weaknesses:**

Strengths :

- The proposed framework gives a precise meaning to procedures that are now frequently used in practice of unrolling (potentially nonsmooth) iterative algorithms programmed in modern autodiff packages (Tensorflow, Pytorch, JAX…) and were shown to work and give good results when optimizing some of their parameters, but without a firm theoretical explanation of why it worked and the derivatives were well defined. In this sense, the work is a valuable theoretical contribution

- The linear convergene of piggyback derivatives is proven for a number of classical algorithms of interest, and is experimentally confirmed through simple examples. A counterexample is given, showing that one must be careful with momentum methods (though I assume that in most practical cases the functions will be regular enough for this not to happen).


Weaknesses :

- I believe the authors should have provided more references to works actually using piggyback derivatives of nonsmooth algorithms in different contexts, to give precide illustrations beyond the toy examples used in the experiments. This would strengthen the impact of the paper and point practical cases for which we now have guarantees.

- The presentation is clear enough to understand the general ideas, but the paper is not exactly self-sufficient. In particular, I believe one must already be very familiar with concepts developed in [12] to understand the tools that must be leveraged. Maybe it would have been nice to include a crash-course exposition to these concepts in the appendices.

---

> ### Author Response · Authors · 2022-07-29
> **Response to Reviewer Sht3**
>
> We thank the reviewer for his positive feedback and provide a detailed response to his comments below:
>
> **I believe the authors should have provided more references to works actually using piggyback derivatives of nonsmooth algorithms in different contexts, to give precise illustrations beyond the toy examples used in the experiments. This would strengthen the impact of the paper and point practical cases for which we now have guarantees.**
>
> Indeed, we will point to more references in the introduction; please do not hesitate to provide references we have missed, this would be of great help.
>
> As for what we are aware of, reference [8] provides direct application to hyperparameter tuning in inverse problem, while the special case of LASSO is analyzed in [7]. Further relevant references include for example
> - Hurault, Leclaire, Papadakis, Gradient step denoiser for convergent plug-and-play, ICLR 2022.
> - Deledalle, C. A., Vaiter, S., Fadili, J., & Peyré, G. (2014). Stein Unbiased GrAdient estimator of the Risk (SUGAR) for multiple parameter selection. SIAM Journal on Imaging Sciences, 7(4), 2448-2487.
> - Ochs, P., Ranftl, R., Brox, T., & Pock, T. (2016). Techniques for gradient-based bilevel optimization with non-smooth lower level problems. Journal of Mathematical Imaging and Vision, 56(2), 175-194.
> - Bertrand, Q., Klopfenstein, Q., Massias, M., Blondel, M., Vaiter, S., Gramfort, Salmon, J. (2022). Implicit differentiation for fast hyperparameter selection in non-smooth convex learning m. Journal of Machine Learning Research, 23(149), 1-43.
>
> **The presentation is clear enough to understand the general ideas, but the paper is not exactly self-sufficient. In particular, I believe one must already be very familiar with concepts developed in [12] to understand the tools that must be leveraged. Maybe it would have been nice to include a crash-course exposition to these concepts in the appendices.**
>
> Indeed this is clearly missing. We will add more material to explain the most relevant aspects of [12], emphasizing intuition and key results. .
>
>
> **Is it possible to find a case of practical interest where the heavy ball algorithm (or other momentum algorithms) will fail ? I feel like, from the example given, than in most cases the cost functions are regular enough for this not to happen, but I would appreciate to be proven wrong.**
>
> For more regular, $C^2$ objective $f$, the derivatives of the heavy ball algorithm will converge under strong convexity. This was detailed in [34] and is actually a consequence of [24]. So indeed, more smoothness provides convergent derivatives. It is true that our example is very specific and does not really correspond to any practical situation. We see this example as a theoretical limitation, related to a very bad alignment, but indeed, this behavior may not be generic, and in particular, convergence of derivatives may occur in most practical settings for the heavy ball. This is indeed an interesting question for which we do not have any strong opinion yet.
>
>
> **How hard would the analysis get in the case where there are multiple fixed points of F ? E.g. when minimizing a nonconvex function ? I understand the analysis would get much more complex, and is probably out of the scope of the paper, but this could have been discussed a bit.**
>
> This is a very relevant question, we have several elements of answer:
> - Assumption 1 may be required to hold only locally (Remark 1), which would allow to relax convexity to some extent (but not uniqueness of the fixed point).
> - Multiple fixed points may cause a major difficulty: the fixed point mapping may become discontinuous with respect to initialization $x_0$. The conservative gradient framework developed in [12] allows to handle locally Lipschitz functions but not discontinuous functions (as integration along absolutely continuous curves cannot hold). This is a major difficulty which would require to take a completely different path. We do not see any direct generalization to this setting.
> - An intermediate case would be to consider multiple fixed points which have a continuous dependency on initial conditions of the algorithm. However verifying such an assumption in practice may be quite hard so that the resulting results would apply only to very specific problems (potentially close to convex). An example of step in this direction is given in (Non-Convex Bilevel Games with Critical Point Selection Maps, Arbel and Mairal, 2022) where Section 4 describes convergence of derivatives of continuous time gradient flow in a setting where implicit differentiation does not apply.

---

### Official Review · Reviewer_8Cko · 2022-07-06

**Rating:** 7
**Confidence:** 4
**Soundness:** 4 excellent
**Presentation:** 3 good
**Contribution:** 4 excellent

**Summary:**

In this paper, the authors consider a possibly non-smooth but structured iterative algorithm which additionally depends on some parameter $\theta$, also in a possibly non-smooth but structured manner; the structure being defined in Assumption 1. They used the tools developed in [1, 2] to differentiate this parametric fixed-point iteration through Chain Rule. This defines a (so-called) piggyback recursion, where the sequence being generated is set-valued. They show that the conservative jacobian of the fixed-point $\bar x (\theta)$ of the algorithm satisfies a set-valued, affine fixed-point equation. They show the convergence of the resulting sequence to the conservative jacobian of the fixed-point of the algorithm for all $\theta$ with respect to the gap metric defined in Section 3.1. They also link the implicit differentiation of the fixed-point $\bar x (\theta)$ by using the conservative calculus in [3] to the limit of the set-valued sequence. They show linear convergence of this sequence under additional Lipschitz-gradient-selection assumption of the algorithm mapping $F$. Finally, the consider instances of the iteartive algorithm to solve a composite optimization problem, namely, Forward-Backward Splitting, Douglas Rachford and ADMM. They put another assumption on the optimization problem (Assumption 2) and show that Assumption 1 is satisfied by the above listed algorithms. They also show the failure of convergence of the derivative sequence for inertial methods.

[1] Jerome Bolte and Edouard Pauwels. Conservative set valued fields, automatic differentiation, stochastic gradient method and deep learning. Mathematical Programming, 2020.

[2] Jerome Bolte and Edouard Pauwels. A mathematical model for automatic differentiation in machine learning. In Conference on Neural Information Processing Systems, Vancouver, Canada, 2020.

[3] Jerome Bolte, Tam Le, Edouard Pauwels, and Antonio Silveti-Falls. Nonsmooth Implicit Differentiation for Machine Learning and Optimization. In Advances in Neural Information Processing Systems, Online, 2021.

**Questions:**

- What do the authors think of the difference in using Automatic and Implicit Differentiation for non-smooth fixed-point iterations and equations respectively in terms of
    - the computational efficiency of the algorithms in their setting, and
    - the correct evaluation of the derivative of the fixed-point (or something "reasonable" when that does not exist) in a more general setting.
- If the answer to the previous question is hat Implicit Differentiation behaves atleast as well as Automatic Differentiation then why even use Automatic Differentiation since Implicit Differentiation is memory efficient.

Minor Typos and other Confusions
- Page 3, Line 94: "...classes convex functions..." -> "...classes of convex functions...".
- Page 3, Line 110: $J_{x_1}$ was not defined.
- Page 4, Line 132, Equation (6): "$[A, B] \in J$" -> "$[A, B] \in \mathcal J$".
- Page 4, Line 145: "...Jacobian of the fixed-point of $F_\theta$" is misleading -> "...Jacobian at the fixed-point of $F_\theta$" is more suitable perhaps.
- Page 5, Line 153: "...this a limit..." -> "...this is a limit...".
- Page 7, Line 220: "...Proposition~1, as well Corrollar~3..." -> Sentence does not look correct gramatically.

**Limitations:**

See Weaknesses. Other than that, in my opinion their work is sound.

**Strengths And Weaknesses:**

Strenghts
- The paper provides a fairly general result for non-smooth iterative algorithms. This is useful for instance in a bilevel framework since their result along with [1, 2] provides theoretical guarantees for correctly evaluating the gradient of the upper level problem by differentiating through the argmin of the lower problem by using Automatic Differentiation.

Weaknesses
- A comment on the difference between the convergence speed of Piggyback iterations for the mentioned optimization algorithms is missing.

---

> ### Author Response · Authors · 2022-07-29
> **Response to 8Cko (2/2)**
>
> **"the correct evaluation of the derivative of the fixed-point (or something "reasonable" when that does not exist) in a more general setting"**
>
> - The set valued map $x \rightrightarrows J_{imp}(x)$ has a closed graph and therefore $gap(J_{imp}(x_k), J_{imp}(\bar{x}))$ will converge to $0$ similarly as $gap(J_{x_k}, J_{fix})$ converges to $0$. We do not have quantitative ways to estimate difference between both quantities.
> - Intuitively $J_{imp}(\bar{x}) \subset J_{fix}$ and the second set may be easier to converge to (this is just a guess).
> - More formally, automatic differentiation ensures that for all $k$, $J_{x_k}$ is a conservative Jacobian for $x_k$. An example of consequence is that for almost all $\theta$, and for all $k$, $J_{x_k}(\theta)$ is equal to the classical Jacobian. No such property holds for $J_{imp}(x_k)$, and beyond its convergence, for a fixed $k$ it does not carry variational meaning. This might have consequences, for example for the Lasso hyperparameter tuning problem with forward backward algorithm (e.g. in [7]), implicit differentiation would work accurately after identification of the support of the solution (active manifold), but before such an identification, nothing can really be said about the meaning of $J_{imp}(x_k)$ while $J_{x_k}$ retains some meaning (it is conservative for $x_k).
>
>
> **If the answer to the previous question is hat Implicit Differentiation behaves at least as well as Automatic Differentiation then why even use Automatic Differentiation since Implicit Differentiation is memory efficient.**
>
> We agree with the remark of the reviewer and we have two comments to make:
> - The previous analysis suggests that for moderate values of $K$ or $m$ the computational time of backward AD is more favorable. For instance,  well-conditioned problem generally yield fast convergence so it is likely that they form a favorable class for fast PB processes.
> - Implicit differentiation does not have a variational meaning for a fixed $K$ while this is the case for automatic differentiation. This could be interesting in a "sketchy calculus" context or when constraint identification plays a significative role.
>
> More speculatively, convergence of automatic differentiation may occur in situations whereas the implicit function theorem does not apply due to lack of invertibility (with a potential dependency on initial conditions). An example in this direction is given in (Non-Convex Bilevel Games with Critical Point Selection Maps, Arbel and Mairal, 2022) where Section 4 describes convergence of derivatives of continuous time gradient flow in a setting where implicit differentiation does not apply.
>
> These are very qualitative and high level responses and we consider that the question raised by the reviewer is essentially open in a nonsmooth context and requires further investigations.
>
>
> **Minor Typos and other Confusions**
>
> Thanks for catching these typos, in particular, we will define the initialization more precisely as also pointed out by reviewer nZjc.

---

> ### Author Response · Authors · 2022-07-30
> **Response to reviewer 8Cko (1/2)**
>
> We thank the reviewer for his positive feedback and his relevant questions.
>
> **A comment on the difference between the convergence speed of Piggyback iterations for the mentioned optimization algorithms is missing.**
>
> Yes definitely, we will add a discussion in Section 4.1 along the following lines:
>
> - Consider for instance a sum composite problem: take a $\mu$ strongly convex $f$ with $L$-Lipschitz gradient, and $g$ convex and consider Forward-Backward recursion (FB) and Douglas-Rachford recursion (DR) (we ignore ADMM which deals with different types of problem and which is anyway an instance of DR). Corollary 3 suggests that the asymptotic linear convergence factor can be chosen as closed at desired to $\sqrt{\rho}$ where $\rho$ is the contraction factor of the algorithm. Actually we suspect that the square root is an artifact and could be removed, but do not have a proof yet.
>
> Therefore one needs to compute the contraction factor of each algorithms. This results in the following, for FB and DR iterations, set $\tau = L/\mu$ the condition number of $f$
> - FB: choosing $\alpha = 2 / (L + \mu)$ results in contraction factor $\rho_{FB} = (\tau - 1) / (\tau + 1)$
> - DR: (see for example [25] proposition 3), choosing $\alpha = 1 / \sqrt{L \mu}$ results in contraction factor $\rho_{DR} = (\sqrt{\tau} - 1) / \sqrt{\tau} + 1) < \rho_{FB}$
>
> Corollary 3 suggests asymptotic linear convergence factor of order $\sqrt{\rho_{FB}}$ for Forward Backward piggy back derivatives and $\sqrt{\rho_{DR}} < \sqrt{\rho_{FB}}$ for Douglas Rachford piggy back derivatives. The obtained rate is asymptotically faster for DR than FB. Note that the limiting conservative gradients need not to be the same.
>
> **What do the authors think of the difference in using Automatic and Implicit Differentiation for non-smooth fixed-point iterations and equations respectively**
>
> This is a very relevant question which was part of our interrogations while writing this manuscript and for which we do not have an answer yet.
>
> **"the computational efficiency of the algorithms in their setting":**
>
> Let us sketch a possible answer. Assume that we perform $K$ iterations of the algorithms. Denote by $cost$ the computational cost, we have $cost(x_K) = O(K cost(F))$. Let us also assume that $cost(J_F) \leq a cost(F)$ for a small constant $a$ (cheap derivative principle). Denote by $c(p,m)$ the cost of $p \times p$ matrix multiplication with $p \times m$ matrix. It is $p^2 \times m$ for the naive multiplication algorithm, and smaller for smarter algorithms, as Strassen's method. We use  the naive algorithm for simplicity.  Denote by $J_{imp}$ the conservative Jacobian obtained by algorithmic differentiation.
>
> - Using the recursion (PB) $cost(J_{x_K}) = O( (1+a) K cost(F) + K p^2 \times m)$ and the memory footprint is of the order of $p\times m$ for the current Jacobian matrix $J_k$, $k = 0,\ldots, K$.
>
> - Using the forward mode JVP in Algorithm 1 $cost(J_{x_K} \dot{\theta}) = O( (1+a) K cost(F) + K p \times m)$ and the memory footprint is of the order of $p$ for the current iterate $x_k$, $k= 0,\ldots, K$. Note that computing $w_K^T J_{x_K}$ using forward mode JVP algorithm essentially requires to compute the full recursion (PB) since $J_{x_K} \dot{\theta}$ will typically "one coordinate" of $w_K^T J_{x_K}$ in the form $w_K^T J_{x_K}\dot{\theta}$.
>
> - Using backward mode VJP in Algorithm 1 $cost(w_K^T J_{x_K}) = O( (1+a)K cost(F) + K p\times m)$ and the memory footprint is of the order of $Kp$ for the $K$ iterates $x_0\ldots, x_K$.
>
> - Using implicit differentiation, $cost(J_{imp}(x_K)) = O((K+a) cost(F) + p^3 + p^2m)$ where the cube is for matrix inversion and the memory footprint is of order $p$ to store the last iterate $x_K$.
>
> This (very rough) list of complexity evaluation suggest that the backward mode costs less time when $Km \leq p^2$, that is the dimension is large with respect to the geometric average $\sqrt{Km}$. It is the case when there is a small number of parameters $m$ or a small number of iterations $K$. This also happens when $K \leq p$, the number of iterations is smaller than the number of dimensions.
> Overall, the backward mode is less memory efficient and could be more time efficient in some regimes, when the number of parameters the number of iterations are small.

---

### Official Review · Reviewer_nZjc · 2022-07-11

**Rating:** 7
**Confidence:** 3
**Soundness:** 3 good
**Presentation:** 3 good
**Contribution:** 3 good

**Summary:**

The authors investigate the differentiation of iterates $x_k(\theta)\in \mathbb{R}^p$ of nonsmooth optimization algorithms with respect to algorithm parameters $\theta \in \mathbb{R}^m$. This task is motivated by the need to differentiate through the argmin of objectives that give rise to nonsmooth algorithmic iterations $x_{k+1} = F(x_k(\theta), \theta)$. Differentiation with respect to algorithmic iterates is frequently used because the argmin itself might for example be incaccessible to explict differentiation or alternative approaches.

An important question is whether Jacobians with respect to algorithmic iterates converge to classical Jacobians at fixed points $\bar x(\theta)$. This has previously been answered for the case of smooth iterations and the authors consider the nonsmooth case in this paper. To that end, they first introduce conservative Jacobians that are set-valued and enable a recursive differentiation framework for nonsmooth iterations. Under assumptions, they show that the defined recursion converges to a nonempty and compact set of fixed points in $\mathbb{R}^{p\times m}$ that is a conservative Jacobian of $\bar x(\theta)$. The limit of conservative Jacobians of $x_k(\theta)$ is therefore equal to a conservative Jacobian at $\bar x(\theta)$. In other words: Limit formation and differentiation can be exchanged here.

Afterwards, the authors also provide a linear convergence result for a specific class of functions (where the iteration is a Lipschitz gradient selection function), they show that their framework is applicable to several splitting algorithms including numerical examples, and they finally provide a concrete counterexample using the heavy-ball algorithm, where the sequence of derivatives diverges although the iterates themselves are linearly convergent.

**Questions:**

Line 77: Should it be $x\in\mathbb{R}^p$ rather than $x\in\mathbb{R}^n$?

Line 89: Should it be $\mathbb{R}^{m\times p}$ rather than $\mathbb{R}^{p\times m}$?

Line 110: How do you initialize $J_{x_0}(\theta)$?

Line 131: Should it be $X \in \mathbb{R}^{p\times m}$ rather than $\mathbb{R}^{p\times n}$?

**Limitations:**

Yes.

**Strengths And Weaknesses:**

I think that this is an interesting and well-written paper. To the best of my knowledge, the presented results for nonsmooth algorithms are novel and also relevant (from a theoretical point of view but also in view of a growing number of use-cases of algorithm unrolling). Moreover, the authors provide rigorous proofs of their results in the appendix. One small weak point is, according to my personal impression, that the paper is partially not self-contained. For example the inputs of algorithmic differentiation in Section 3.3 could at least be briefly explained.

---

> ### Author Response · Authors · 2022-07-29
> **Response to reviewer nZjc**
>
> We thank the reviewer for his positive assessment and critical feedback. A point by point reply is found below.
>
> **One small weak point is, according to my personal impression, that the paper is partially not self-contained. For example the inputs of algorithmic differentiation in Section 3.3 could at least be briefly explained.**
>
> Thanks for pointing this out, this section is indeed a bit fast. We will add a more precise description of input quantities: in a compositional model $\dot{\theta}$ is the gradient of an inner functions $\theta(\lambda)$ controlling algorithm parameters $\theta$ with another variable real variable $\lambda$, for example an hyper parameter. The goal is to  combine $\frac{\partial x_k(\theta)}{\partial \theta}$ and $\frac{\partial \theta}{\partial \lambda}$ in a forward path to implement the chain rule and obtain the total derivative $\frac{\partial x_k(\theta(\lambda))}{\partial \lambda}$. On the other hand, in a compositional model, $\bar{w}_k$ is typically the gradient of an outer loss functions $\ell$ evaluated at $x_k(\theta)$. In this case the goal is to combine derivatives of iterates $\frac{\partial x_k}{\partial \theta}$ with $\bar{w}_k = \frac{\partial \ell(x_k)}{\partial x_k}$ in a backward path to output the total derivative $\frac{\partial l(x_k(\theta))}{\partial \theta}$.
>
> Section 3.3 will be made more natural and intuitive using the above discussion.
>
> **How do you initialize $J_{x_0}(\theta)$?**
>
> This is indeed missing, we will add the following sentence after (PB) on page 3 and all Corollaries of Theorem 2
>
> $J_{x_0} \colon \mathbb{R}^m \rightrightarrows \mathbb{R}^{p \times m}$ is any conservative Jacobian of $\theta \mapsto x_0(\theta)$.
>
> Note that this is not required for the convergence to hold as the initial condition is progressively "forgotten" so that the limit does not depend on it. However it is important to precise what is the initialization and a conservative Jacobian is what was considered implicitly.
>
> **Additional questions:**
>
> Thanks for catching the remaining typos, they will be fixed.

---

### Official Review · Reviewer_F1Pw · 2022-07-14

**Rating:** 7
**Confidence:** 3
**Soundness:** 4 excellent
**Presentation:** 3 good
**Contribution:** 3 good

**Summary:**

This paper introduced a theory for the piggyback iteration of nonsmooth functions. While existing asymptotic analysis for the piggyback AD procedure only works for twice continuously differentiable functions, with the conservative derivative framework, this paper provided the asymptotic convergence results for nonsmooth path-differentiable functions under a nonexpansive condition (in Assumption 1). They formally defined the notion of general Clarke Jacobian for piggyback and its fixed point, with which the convergence of an infinite piggyback iteration is proved in Corollary 1. For mapping with a Lipschitz gradient selection, they proved a non-asymptotic convergence rate. They also showed applications by applying the piggyback iterations to various methods for convex problems.

**Questions:**

See above.

**Limitations:**

Yes.

**Strengths And Weaknesses:**

The paper is well-written with a clear introduction to various new and existing notions in the conservative derivative framework. The problem studies are timely and important. The theoretical results are nontrivial and demonstrate wide extensibility, which also suggests important applications. The failure example discussed in Section 5 is interesting. It shows that even functions with a Lipschitz gradient cannot ensure the convergence of the piggyback procedure. For assumption 1, how does this assumption compare with these in the smooth cases? For example, in [37], they assume the Hessian at optimum is positive definite and they use the gradient descent method. So it would be better to discuss the strength of assumption 1 in various situations where we assume the function is C^2 and fixed the algorithm. Is Assumption 1 for the nonsmooth procedure stronger or weaker than that in existing papers? Some minor points:

L131: "X\in R^{p \times n}" -> "X\in R^{p \times m}"
L153: "this a" -> "this is a"

---

> ### Author Response · Authors · 2022-07-29
> **Response to Reviewer F1Pw**
>
> We thank the reviewer for his positive assessment and critical feedback. A point by point reply is found below.
>
> **For assumption 1, how does this assumption compare with these in the smooth cases?  For example, in [37], they assume the Hessian at optimum is positive definite and they use the gradient descent method. So it would be better to discuss the strength of assumption 1 in various situations where we assume the function is C^2 and fixed the algorithm. Is Assumption 1 for the nonsmooth procedure stronger or weaker than that in existing papers?**
>
> In the smooth case a natural candidate for being a conservative Jacobian is the classical Jacobian and our assumption is very close to what is used in the smooth case in [24,26,15] and more recently [37,34], but there are differences motivated by the surprisingly complex behavior of spectral radii of family of matrices.
>
> _The case of gradient descent:_ In [37], local smoothness and strong convexity ensure that the smooth counterpart of Assumption 1 holds locally for the gradient descent mapping. One difference is that the assumption in [37] is local, but this is also compatible with our analysis as explained in Remark 1, independently of smoothness. All in all, for gradient descent on a $C^2$ convex function, Assumption 1 with the  classical Jacobian (and its local version, Remark 1) exactly corresponds to the setting of [37]. This will be made clearer after assumption 1.
>
> _Beyond gradient descent:_ Our assumption is actually stronger than what is required in existing literature on abstract iterative schemes. Indeed in the smooth case, for general algorithm $F$ it is shown in [24] that a bound on the spectral radius of the Jacobian is sufficient to ensure convergence. Spectral radius and operator norm are the same in the _symmetric case_ (the Jacobian of gradient descent mapping is a Hessian), but not in general (e.g. Jacobian of heavy ball mapping). On the other hand, in the nonsmooth setting, our heavy-ball counterexample shows that uniform spectral radius bound on Clarke Jacobian is not enough to ensure convergence of derivatives. The intrinsic reason for this discrepancy is as follows : for a single matrix $M$ with spectral radius $\rho$, and any $\epsilon > 0$, there exist a scalar product and induced norm $\Omega$ such that $M$ is $\rho+\epsilon$ Lipschitz. So up to linear a change of variables, spectral radius and operator norm are almost the same for a single matrix, and the analysis carries through in the smooth case because the (usual) Jacobian is a single matrix. However, for two matrices $M_1$, $M_2$ with respective spectral radius $\leq \rho$, such a norm, ensuring  both $M_1$ and $M_2$ to be $\rho+\epsilon$ Lipschitz, may not exist. Actually the existence of such a norm  strongly relates to undecidable problems (see "The boundedness of all products of a pair of matrices is undecidable", Blondel and Tsitsiklis, systems and control letters 2000)!
>
> This is the reason why Assumption 1 is on operator norm and not on spectral radius. This is sharp, as illustrated by the heavy ball example for which derivatives converge in the C2 case (e.g. [37]), but not for less regular cases even when the spectral radii are <1. This discussion will also be reflected after Assumption 1 and in Section 5.
>
> **Some minor points: L131: ... L153: ...**
> Thanks for catching these typos, they will be corrected.

---

### Meta-Review · Area_Chair_T8uN · 2022-08-25

**Recommendation:** Accept
**Confidence:** Certain

**Metareview:**

The reviewers agreed that the paper has solid and novel technical contributions. Nevertheless, please consider elaborating more on the background of the techniques used in the revision, so that the paper is more self-contained.

**Award:**

No

---

### Decision · Program_Chairs · 2022-09-14

Accept